# The *ex planta* signal activity of a *Medicago* ribosomal uL2 protein suggests a moonlighting role in controlling secondary rhizobial infection

Fernando Sorroche[1]\*, Violette Morales[2], Saïda Mouffok[1], Carole Pichereaux[3,4], A. Marie Garnerone[1], Lan Zou[1], Badrish Soni[1¤a], Marie-Anne Carpéné[5¤b], Audrey Gargaros[4¤c], Fabienne Maillet[1], Odile Burlet-Schiltz[4], Verena Poinsot[5], Patrice Polard[2], Clare Gough[1], Jacques Batut[1]\*

1 Laboratoire des Interactions Plantes Microorganismes (LIPM), INRAE, CNRS, Castanet-Tolosan, France, 2 Laboratoire de Microbiologie et de Génétique Moléculaires, UMR5100, Centre de Biologie Intégrative (CBI), Centre National de la Recherche Scientifique (CNRS), Université de Toulouse, UPS, Toulouse, France, 3 Fédération de Recherche (FR3450), Agrobiosciences, Interactions et Biodiversité (AIB), CNRS, Toulouse, France, 4 Institut de Pharmacologie et de Biologie Structurale (IPBS), Université de Toulouse UPS, CNRS, Toulouse, France, 5 I2MC, Université de Toulouse UPS, INSERM, CNRS, Toulouse, France

¤a Current address: Reliance Research and Development Centre, Reliance Industries Limited, Ghansoli, Navi Mumbai, India
¤b Current address: Deltalab-SMT, Carcassonne, France
¤c Current address: Evotec SAS, Toulouse, France
\* ferguisor@gmail.com (FS); Jacques.Batut@inrae.fr (JB)

**Data Availability Statement:** The annotation of 6 M. truncatula RPuL2 proteins has been deposited in GenBank: MtrunA17CPg0492511.2 MT965675

## Abstract

We recently described a regulatory loop, which we termed autoregulation of infection (AOI), by which *Sinorhizobium meliloti*, a *Medicago* endosymbiont, downregulates the root susceptibility to secondary infection events *via* ethylene. AOI is initially triggered by so-far unidentified *Medicago* nodule signals named signal 1 and signal 1' whose transduction in bacteroids requires the *S. meliloti* outer-membrane-associated NsrA receptor protein and the cognate inner-membrane-associated adenylate cyclases, CyaK and CyaD1/D2, respectively. Here, we report on advances in signal 1 identification. Signal 1 activity is widespread as we robustly detected it in *Medicago* nodule extracts as well as in yeast and bacteria cell extracts. Biochemical analyses indicated a peptidic nature for signal 1 and, together with proteomic analyses, a universally conserved *Medicago* ribosomal protein of the uL2 family was identified as a candidate signal 1. Specifically, MtRPuL2A (MtrunA17Chr7g0247311) displays a strong signal activity that requires *S. meliloti* NsrA and CyaK, as endogenous signal 1. We have shown that MtRPuL2A is active in signaling only in a non-ribosomal form. A *Medicago truncatula* mutant in the major symbiotic transcriptional regulator MtNF-YA1 lacked most signal 1 activity, suggesting that signal 1 is under developmental control. Altogether, our results point to the MtRPuL2A ribosomal protein as the candidate for signal 1. Based on the *Mtnf-ya1* mutant, we suggest a link between root infectiveness and nodule development. We discuss our findings in the context of ribosomal protein moonlighting.

MtrunA17Chr3g0090931.2 MT965676
MtrunA17Chr3g0094621.2 MT965677
MtrunA17Chr4g0016021.2 MT965678
MtrunA17Chr4g0017201.2 MT965679
MtrunA17Chr4g0024461.2 MT965680. All other
relevant data are within the manuscript and its
Supporting Information files.

**Funding:** FS was supported by a Post-doctoral
AGREENSKILLS fellowship and a ANR (ANR-15-
CE20-0004-01) post-doctoral fellowship. LZ was
supported by a CSC PhD scholarship. BS was
supported by a INRA SPE post-doctoral fellowship.
MAC was supported by a PhD fellowship from the
French Ministère de l'Enseignement supérieur et de
la Recherche. AG. This work was funded in part by
the ANR "RhizocAMP" (ANR-10-BLAN-1719), the
ANR "AOI" (ANR-15-CE20-0004-01) and the Pôle
de Compétitivité "Agri Sud Ouest Innovation". This
work is part of the "Laboratoire d'Excellence"
(LABEX) entitled TULIP (ANR-10-LABX-41). The
proteomic part of this project was supported in
part by the Région Occitanie, European funds
(Fonds Européens de DEveloppement Régional,
FEDER), Toulouse Métropole, and by the French
Ministry of Research with the Investissement
d'Avenir Infrastructures Nationales en Biologie et
Santé program (ProFI, Proteomics French
Infrastructure project, ANR-10-INBS-08). The
funders had no role in study design, data collection
and analysis, decision to publish, or preparation of
the manuscript.

**Competing interests:** The authors have declared
that no competing interests exist

## Introduction

The establishment and maintenance of symbiotic relationships is tightly regulated by bidirectional signaling between host and symbiont. Given the critical importance of these signaling events, the molecular identification of the underlying signals is a major challenge in the field. In one of the most significant and widespread beneficial plant-microbe interaction, legume plants get their nitrogen supply from a symbiotic relationship with nitrogen-fixing soil bacteria called rhizobia that reduce atmospheric nitrogen to ammonia for them [1–3]. Nitrogen fixation takes place in nodules, which are dedicated mixed organs that rhizobia elicit on the roots of compatible plants and that they colonize intracellularly. Inside nodules, endosymbiotic rhizobia called bacteroids trade fixed nitrogen in exchange of plant photosynthates and a protected niche.

Signal exchange between the two symbionts takes place all along this long-lasting, mutualistic, interaction [4]. Both bacterial and plant signals contribute to the development of a functional nodule. Root-secreted flavonoids trigger synthesis of lipo-chitooligosaccharides (LCOs, also called Nod factors) by rhizobia in the rhizosphere. Nod factors then simultaneously trigger nodule formation in the root cortex and the formation of specialized infection structures called infection threads (ITs) in the epidermis ([5] for a recent review). Bacterial exopolysaccharides also contribute to IT formation and enhance bacterial survival [6–8]. Inside the nodules, hypoxia triggers bacteroid differentiation and nitrogen fixation [9, 10]. In the IRLC (Inverted Repeat-Lacking Clade) clade of legumes, nodule cysteine-rich (NCR) plant peptides further control bacteroid differentiation [11–13].

Autoregulatory loops contribute to the maintenance of mutualism in two ways. The autoregulation of nodulation (AON) process keeps nodule number in balance with the plant nitrogen needs and carbon availability. During AON, plant CLE peptides synthesized in nodules migrate to the shoot and trigger a feedback loop that decreases the root susceptibility to nodulation [14]. We recently discovered another mechanism that specifically connects new IT formation to nodule number in the *Medicago-S. meliloti* symbiosis [15]. This loop, which we termed autoregulation of infection (AOI), involves a novel type of signal exchange between the two symbionts. In the *Medicago* nodule, two so-far unknown plant signals called signal 1 and 1' trigger cAMP signaling in bacteroids [16, 17]. In turn, bacteroids boost the production of ethylene, a known inhibitor of infection, by the plant thus decreasing the root susceptibility to secondary infection by rhizospheric bacteria [15]. Signal 1 and signal 1' perception require the NsrA outer membrane receptor protein in bacteroids [17]. Bacterial signal transduction involves specific, inner-membrane associated, adenylate cyclases, CyaK for signal 1 and CyaD1/D2 for signal 1' [16, 17].

We previously established a bioassay in which *S. meliloti* free-living bacteria carrying the cAMP-dependent *smc02178-lacZ* reporter fusion detected the presence of signal 1 in *Medicago* nodule crude extracts, as well as in shoots [16]. *Lotus* and pea nodules as well as rice shoot extracts also displayed signal 1-like activity, suggesting that signal 1 is a widespread molecule in plants (16). Overexpressing the *nsrA* bacterial receptor gene in *S. meliloti* led to a more sensitive bioassay for signal 1 [17].

Here, we show that signal 1 activity is further widespread, from plants to bacteria. Accordingly, we identified by biochemical approaches a universally conserved *Medicago* ribosomal protein of the uL2 family as the best candidate signal 1. We identified two specific isoforms of MtRPuL2, MtRPuL2A and MtRPuL2B, in nodules. Purified MtRPuL2A (MtrunA17Chr7g0247311) displays a strong signal activity in our *ex planta* bioassay that requires *S. meliloti* NsrA and CyaK, as endogenous signal 1 in nodules. Strikingly, MtRPuL2A was active in signaling only in a non-ribosomal form. In addition, we have found that a *Medicago truncatula* mutant in the major symbiotic

transcriptional regulator MtNF-YA1 lacked signal 1 activity. Altogether, our results suggest that a uL2 ribosomal protein has a moonlighting signaling function in symbiosis, possibly linking root infectiveness to nodule development.

# Results

## Signal 1 is widespread and protease sensitive

To explore further the pervasiveness of signal 1, we assayed for its presence in bacteria including *Escherichia coli* and S. *meliloti* and in the yeast *Saccharomyces cerevisiae*. We used a *S. meliloti* strain (GMI12052, S4 Table) that overproduces the NsrA receptor protein as a reporter strain for signal activity. We found that crude cell extracts of free-living cultures of these microorganisms displayed signal activity comparable to that of *Medicago* nodule extracts. Furthermore, signal transduction required *S. meliloti* CyaK, the cognate adenylate cyclase for signal 1 (Fig 1A).

The signal activities of both *Medicago* nodule (Fig 1B) and *E. coli* (Fig 1C) crude extracts decreased rapidly upon treatment with proteinase K suggesting a peptidic nature for signal 1. The signal activity in *E. coli* cells thus mimicked the signal 1 activity of *Medicago* nodules both in terms of CyaK-dependency and protease sensitivity. Since *Medicago* nodules are cumbersome to harvest in amounts compatible with biochemical analyses, we used *E. coli* cultures as a

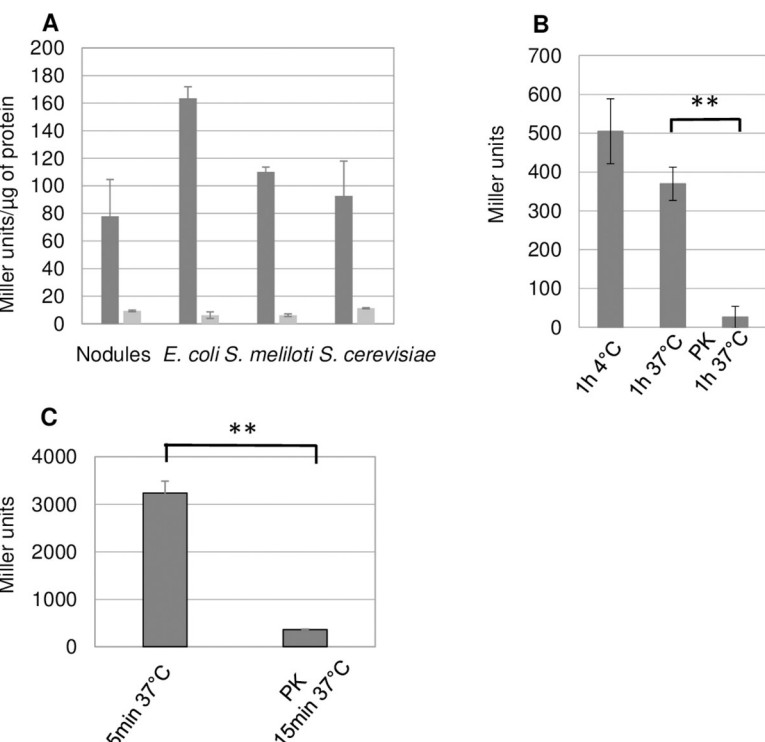

**Fig 1. Signal 1 ubiquity and sensitivity to protease treatment. Panel A:** Crude extracts of *Medicago sativa* nodules, *E. coli* (DH5α), *S. meliloti* (Rm1021) and *S. cerevisiae* (BY4741) were tested for signal activity in *S. meliloti* reporter strains GMI12052 (wt, black) and GMI12071 (*cyaK*,grey). Signal activities are expressed as Miller Units per μg of protein in the crude extracts to ease comparison between crude extracts. Data show the mean values of 3 independent biological repeats. We observed no statistically significant difference in the signal activity of the different crude extracts. **Panel B:** *Medicago n*odule extracts (50 mg) were incubated in the absence or presence of proteinase K (PK), as indicated, before testing their signal activity. P-value 0.0069, t-test, n = 3. **Panel C:** Signal activity of 100 μl of *E. coli* crude extracts untreated or treated with immobilized-PK for 15 min at 37˚C. P-value 0.0023, t-test, n = 3.

source material to attempt signal 1 purification using standard protein purification procedures.

### *E. coli* ribosomal protein RPuL2 (RplB) has signal 1 activity

Signal activity from *E. coli* (DH5α) crude extracts was tracked along five protein purification steps (see S1 Fig and materials and methods). Mass spectrometry analysis of one of the most active fractions eluted from the last Heparin-Sepharose column (B6 in S1 Fig) led to the identification with high confidence of 59 proteins (S1 Table). A survey of the signal activity of cell crude extracts of 30 corresponding mutants available in the *E. coli* Keio collection (https://cgsc.biology.yale.edu/KeioList.php) did not reveal any significant difference as compared to wild-type (S1 Table). Instead, we found that a commercial preparation of *E. coli* topoisomerase I (TopA, Promega) had a significant signal activity. The corresponding mutant did not exist in the Keio collection, as expected since *top1* is an essential gene in *E. coli*. To validate this observation, we therefore extensively purified an amino-terminal His6-tagged version of *E. coli* TopA on Nickel and Heparine-sepharose columns (S1 Fig). We found that the signal activity did not co-elute with the TopA protein itself but with a co-purifying protein (S1 Fig) that we identified by mass spectrometry as being RplB (S1 Table).

RplB (UniProt KB–P60422) is the largest (273 amino acids) ribosomal protein of the large subunit of ribosomes that is essential for ribosome assembly and protein synthesis. RplB is a universal protein that has different names in different organisms. In this article, we have adopted the nomenclature proposed by Ban *et al.* [18] for unifying ribosomal-protein (RP) naming in bacteria, eukaryotes and archaea. In this nomenclature, RplB and homologous eukaryotic L8 proteins have been renamed RPuL2, "u" standing for universal.

We purified a carboxy-terminal strep-tagged version of the RPuL2 (RplB) protein from *E. coli* crude extracts on a Strep-Tactin® column followed by heparine and size exclusion chromatography (Fig 2A). Highly pure fractions containing tagged *E. coli* RPuL2 protein displayed high signal activity (Fig 2B) whose perception in the *S. meliloti* bioassay required both *cyaK* and *nsrA*, as for nodule signal 1 (Fig 2C). Negative controls included the strep-tag peptide, an unrelated (SMc02178) *S. meliloti* strep-tagged protein and a mock (empty vector) purification assay (S2 Fig).

*E. coli* RPuL2 is made up of two domains: an amino-terminal RNA-binding domain (position 1–121) and a highly-conserved, multi-functional, carboxy-terminal domain (122–273) [19]. We cloned and over-expressed the two domains separately as strep-tagged proteins. Both displayed similar signal activities (S3 Fig), in the same range as the full–length RPuL2 protein (Fig 2B). Thus, the signal activity did not relate to a specific functional domain of the protein. These results were consistent with earlier reports that non-ribosomal *E. coli* RPuL2 is a naturally unfolded and intrinsically disordered protein under physiological conditions [20, 21].

RPuL2 proteins carry natural Si-tags at both ends of the protein [22–24], that confer on them a tight binding to silica matrices. Accordingly, the signal activity of a highly purified *E. coli* RPuL2-strep-tag protein was almost completely depleted following chromatography on a fiberglass column (Fig 2D).

Altogether, these results indicated that the *E. coli* RPuL2 protein had signal activity, whose perception by *S. meliloti* required *cyaK* and *nsrA*, as for the genuine *Medicago* signal 1 in nodules.

### A *Medicago* RPuL2 protein as candidate signal 1

Depletion assays indicated that *ca.* 60% of the signal activity present in *Medicago* (A17 wild-type accession) nodule extracts was trapped on a fiberglass column (S4 Fig), suggesting that a

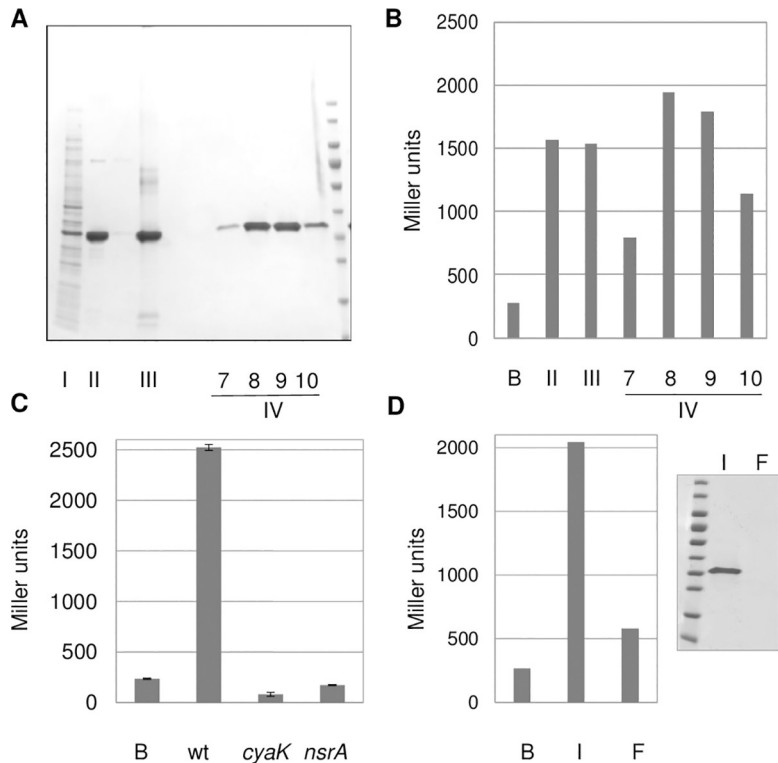

**Fig 2. Signal activity of purified *E. coli* RPuL2 (RplB). Panel A**: SDS-PAGE analysis of purified fractions. I: *E. coli* crude extract of a Strep-tag® overexpressing strain, II: Pool of fractions from Strep-Tactin® resin, III: Pool of fractions from heparine column. IV: Fractions (7–10) after size exclusion column. Wt reporter strain is GMI12052. **Panel B**: signal activity of corresponding fractions diluted as follows: II 60 fold, III 30 fold; IV 10 fold. B control buffer. **Panel C**: Purified RPuL2-strep-tag ® signal activity requires *S. meliloti cyaK* (GMI12071) and *nsrA* (GMI12072). Data show the mean of duplicates with Standard Error. **Panel D**: Fiberglass assay. Signal activity and western-blot (anti-strep tag antibody) monitoring of purified RPuL2-Strep-tag® protein (1μg) before (input, I) and after (flow-through, F) chromatography on fiberglass. B control buffer. Panels B and D feature the results of a single typical experiment.

substantial part of signal 1 activity in nodule extracts was indeed associated with a RPuL2-like protein.

Typically in plants, there are 2 to 4 nuclear-encoded RPuL2 genes coding for highly similar cytosolic proteins, a chloroplast RPuL2 protein encoded by the chloroplast genome, and a mitochondrial protein that is encoded by nuclear and/or mitochondrial genes (when there are two genes they code separately for the amino- and carboxy-terminal parts of the same protein). In addition to these, in the *M. truncatula* genome (version 5, https://medicago.toulouse.inra. fr/MtrunA17r5.0-ANR/), we found 7 additional genes encoding 5 full and 2 truncated proteins highly homologous to the chloroplastic RPuL2 protein (Fig 3 and materials and methods). The potential orthologues of yeast and human RPuL2 proteins were the MtrunA17Chr7g0247311 and MtrunA17Chr5g0405281 proteins that share 97% amino acid identity between them (Fig 3). We therefore named these proteins MtRPuL2A and MtRPuL2B, respectively.

We analyzed by mass spectrometry the protein content of *M.* truncatula A17 nodule extracts after elution from a fiberglass column (see materials and methods). 321 proteins were validated with high confidence from two independent biological replicates (S2 Table). Peptides corresponding to the MtRPuL2A (MtrunA17Chr7g0247311) and/or MtRPuL2B (MtrunA17Chr5g0405281) paralogous proteins were among the most abundantly detected proteins. 3 peptides specific to the MtRPuL2A protein were identified by the mass spectrometry analysis

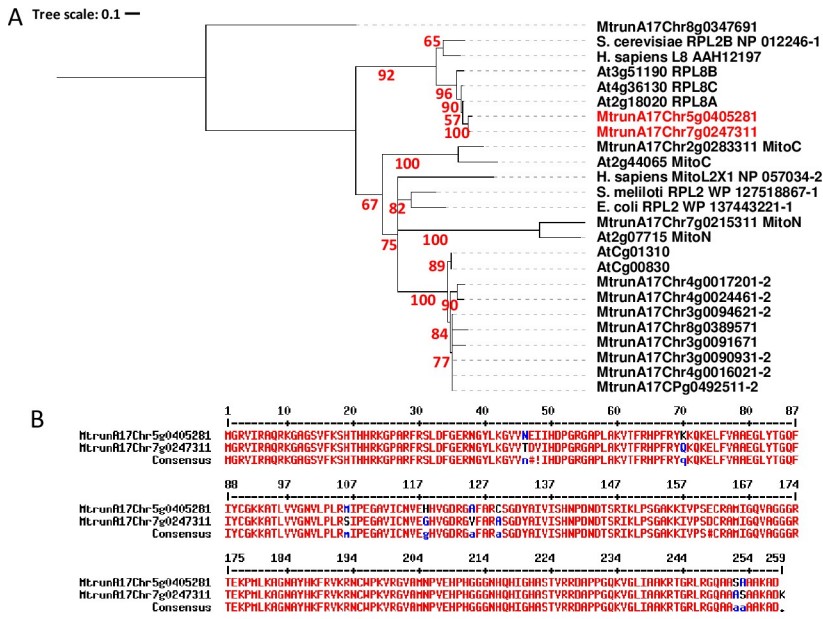

**Fig 3. The MtRPuL2 protein family. Panel A:** phylogenetic tree of RPuL2 protein sequences from *Medicago truncatula* (MtrunA17), *Arabidopsis thaliana* (At), *Homo sapiens*, *Escherichia coli*, *Sinorhizobium meliloti* and *Saccharomyces cerevisiae*. "MitoC" and "MitoN" refer to the C and N-terminal parts of mitochondrial proteins, respectively. "CP" and "C" refers to chloroplastic proteins. Original nomenclatures are kept (L2, L8). MtrunA17Chr7g0247311 (MtRPuL2A) and MtrunA17Chr5g0405281 (MtRPuL2B) proteins are in red. Branches with support values less than 0.5 were collapsed. The MtrunA17Chr8g0347691 RPuL5 protein used for specificity (Fig 5), was considered as an outgroup to root the tree. See methods for details. **Panel B**: sequence alignment of the MtrunA17Chr7g0247311 and MtrunA17Chr5g0405281 proteins (http://multalin.toulouse.inra.fr/multalin).

whereas no peptide specific to the MtRPuL2B protein was validated (S2 Table). Noteworthy, no other protein of the MtRPuL2 family was validated either. 108 *S. meliloti* proteins were detected in the same proteomic analysis but not the *S. meliloti* RPuL2 *(*RplB) protein (S2 Table). MtRPuL2A was thus the best candidate signal 1 molecule in the nodule extract.

As deduced from data publicly available on the *M. truncatula* GeneAtlas (https://mtgea. noble.org/v3/) [25, 26], the *MtRPuL2A* and *MtRPuL2B* genes show very similar patterns of expression in diverse symbiotic and non-symbiotic organs and conditions tested, with a higher expression level for the *MtRPuL2A* gene (S5 Fig). In mature nodules, laser dissection experiments [27] showed a higher expression of the two genes in the apical part of the nodule including the proximal and distal infection zone (ZII) and less expression in the fixation zone (ZIII) of the nodule (S5 Fig).

## Free MtRPuL2A protein has signal 1 activity

We over-produced a carboxy-terminal strep-tagged version of the MtRPuL2A protein in *E. coli* (see materials and methods). Overproduction of the MtRPuL2A-strep protein in *E. coli* could only be achieved at low temperature (16˚C) and, in two independent purification assays, the protein co-purified with a protein that we identified as the *E. coli* GroEL chaperone by Western-blot analysis (Fig 4A). GroEL is a promiscuous chaperone typically associated with misfolded proteins [28]. Purified MtRPuL2A protein displayed high *cyaK*- and *nsrA*-dependent signal activity (Fig 4B) that was prone to fiberglass binding (Fig 4C). The specific activity of the purified MtRPuL2A-strep protein was *ca*. 3-fold lower than that of *E. coli* RPuL2 (*ca* $10^5$ Miller units/μM) assessed in independent purification assays, possibly because of the poor solubility/stability of the protein.

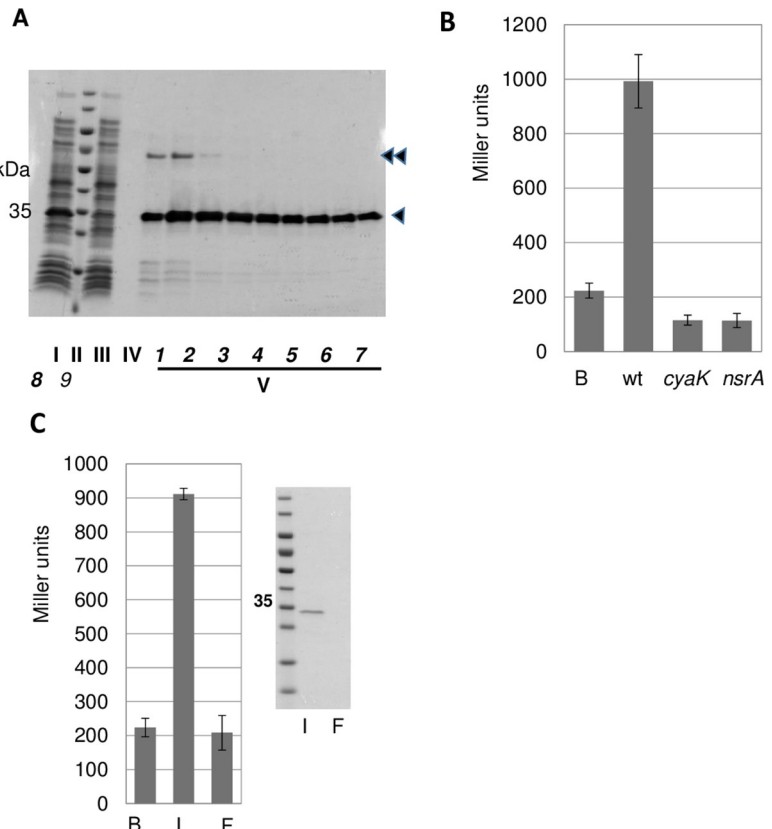

**Fig 4. Purified MtRPuL2A-Strep-tag® has signal activity. Panel A:** SDS-PAGE monitoring of MtRPuL2A-Strep-tag® purification. I: *E. coli* crude extract of a MtRPuL2A-Streptag®overexpressing strain; II-Molecular weight marker; III-Flow-through of the Strep-Tactin® column; IV-Last wash step of the Strep-Tactin® column; V-fractions obtained after elution with 5mM D-desthiobiotin. The simple and the double arrowheads point to the MtRPuL2A-Streptag protein and the *E. coli* GroEL protein, respectively. **Panel B**: activity of pooled purified fractions (V6 to V9; 4-fold dilution) after the Strep-Tactin® column in wild type (GMI12052), *cyaK* (GMI12071) and *nsrA* (GMI12072) reporter strains. B buffer control. **Panel C**: Fiberglass depletion assay of the strep-Tactin purified MtRPuL2A-Streptag protein (7 μg). The signal activities of the input (I) and flow-through (F) fractions of a fiberglass column on a wt *S. meliloti* reporter strain are shown. B buffer control. Right: Western blot control using an anti-strep-tag antibody. Activities are the mean of two independent experiments. Error bars feature SE.

To assess specificity, we purified another ribosomal protein, MtrunA17Chr8g0347691, whose homologue in vertebrates (RPuL5) has a demonstrated extra-ribosomal activity [29, 30]. This protein was detected among the *Medicago* proteins binding fiberglass (S2 Table). A MtrunA17Chr8g0347691-strep-tagged protein purified from *E. coli* had a low specific activity although it purified easily without any associated chaperone (Fig 5). As RPs are very basic proteins (pI 11.1 for MtRPuL2A/MtRPuL2B *vs* pI 9.45 for MtrunA17Chr8g0347691), we tested other cationic compounds for signal activity. A cocktail of 4 different *Medicago* NCR basic peptides (NCR035, NCR055, NCR247, NCR355, pI from 8.46 to 11.53), a histone complex (H2A, H2B, H3, H4; pI 11.4) and the cationic polypeptidic antibiotic Polymyxin B displayed no signal activity (Fig 5) indicating that non-specific electrostatic interactions are not responsible for signal activity. Altogether, these data validated MtRPuL2A as the best candidate signal 1 molecule.

Since RPuL2 proteins are usually components of the large subunit of ribosomes, we purified *M. truncatula* nodule ribosomes by strong anion exchange monolith chromatography [31, 32] using CIMmultus columns™ (BIA separations Inc). Purified ribosomes from *Medicago* A17

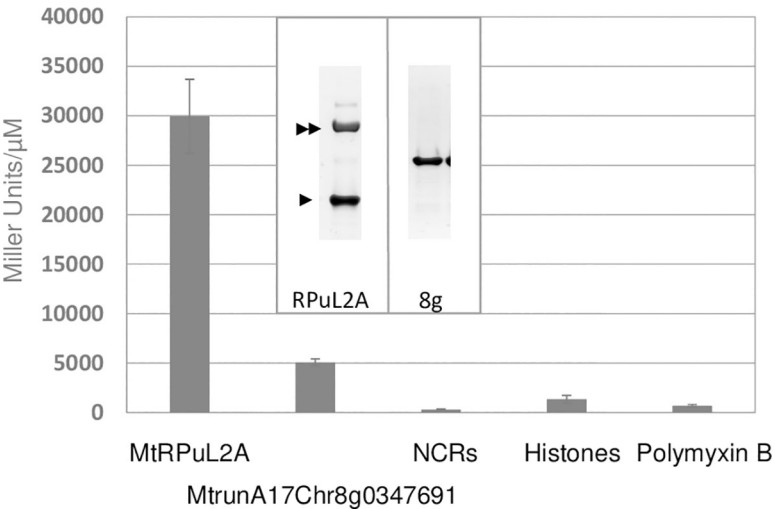

**Fig 5. Specific activities of purified proteins and compounds.** The insert shows SDS-PAGE images of purified protein preparations. The simple and double arrowheads point to the MtRPuL2A and the *E. coli* GroEL proteins, respectively. Activities for the MtRPuL2A and MtrunA17Chr8g0347691 proteins are the mean of two independent purification experiments. n = 3 for NCRs, histones and Polymyxin B. Error bars feature SE. Reporter strain is GMI12052 (wt).

nodules, as well as a commercial preparation of *E. coli* ribosomes (NEB P0763S), displayed no or little signal activity, in contrast to RNAse A-dissociated ribosomes (Fig 6). Mass spectrometry analysis of purified ribosomes from *M. sativa* bacteria-free NAR (Nodulation in the Absence of Rhizobia) nodules confirmed the presence of the MtRPuL2A protein in nodule ribosomes and of the MtRPuL2B protein as well, although with less confidence (S3 Table).

The fact that purified ribosomes did not display any signal activity without RNAse A treatment excluded a spontaneous dissociation of the ribosomes during the bioassay as a source of activity. Furthermore, we found that the RNAse-A treatment of *Medicago* A17 (wt) or *Mtnf-ya1* (see below) nodule extracts did not increase signal 1 activity (S6 Fig, see discussion).

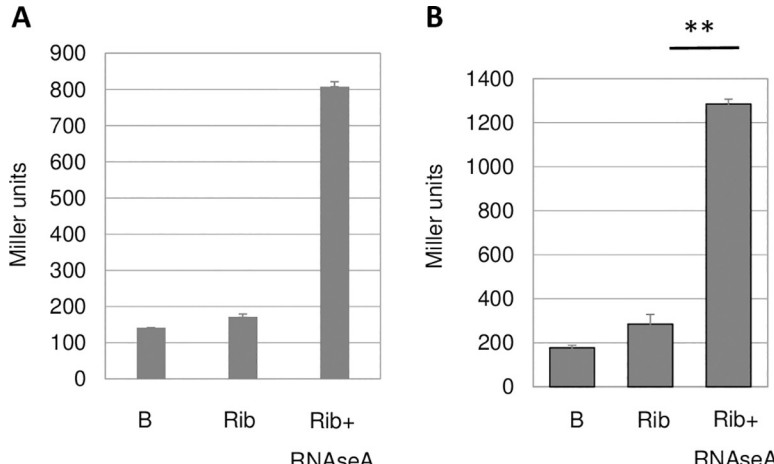

**Fig 6. Free RPuL2 has signal activity.** RNAseA-promoted dissociation of purified ribosomes (Rib) frees signal activity. Reporter strain is GMI12052 (wt). **Panel A:** *Medicago* A17 nodule ribosomes extracted from 100 mg (FW) of nodules. B control buffer. The mean of two experiments is shown. **Panel B:** Signal activity of purified *E. coli* ribosomes (NEB P0763S, 0.5mg). P-value 0.0014, t-test, n = 3.

Conversely, crude extract preparations in the presence of a cocktail of RNAse inhibitors did not markedly affect activity (p-value 0.046) (S6 Fig). Therefore, signal 1 activity did not originate from the spontaneous dissociation of ribosomes during the crude extract preparation either. Instead, these data suggested that free MtRPuL2A protein preexisted physiologically in nodules.

It was shown before that association of *E. coli* RPuL2 with the Hsp90 chaperone stabilizes the free protein by preventing its degradation by the proteasome [33]. Noteworthy, a *Medicago* Hsp90 protein was among the most abundant proteins detected in the nodule protein fraction binding fiberglass (S2 Table), thus providing circumstantial evidence for the presence of non-ribosomal MtRPuL2 protein under physiological conditions in nodules.

### A *Medicago truncatula MtNF-YA1* symbiotic mutant lacks signal 1 activity

No transposon insertion in the *MtRPuL2A* gene was available in a large Tnt1 *Medicago* mutant library (https://medicago-mutant.noble.org/mutant/). In *Arabidopsis*, a At2g18020 mutant (*emb 2296*, *AtRPL8A*) was embryo-defective [34]. Since At2g18020 is one of the 3 cytosolic RPuL2 proteins in *Arabidopsis* (Fig 3), it is possible that mutations in the *MtRPuL2A* gene lead to similar defects. We therefore looked for *Medicago* symbiotic mutants displaying an altered signal 1 activity in nodule crude extracts. The *Mtnf-ya1.1* null mutant was particularly attractive to us as it shows a hyper-infection phenotype [35], possibly indicative of a defective AOI. MtNF-YA1 is a major transcriptional regulator of nodule development whose inactivation stops nodule development prematurely, before the formation of a persistent meristem [35, 36]. *Mtnf-ya1* nodules are small, partially infected (Fig 7A) and fix nitrogen at a very low level [35]. We found that *Mtnf-ya1* nodule extracts had very low signal 1 activity as compared to *M. truncatula* A17 (wt) nodules (Fig 7B). This suggests that the abundance (or activity) of the non-ribosomal MtRPuL2A protein fraction is regulated during nodule development in a NF-YA1-dependent process. Yet, a comparative Western blot analysis of the MtA17 and *MtNF-YA1* nodule extracts did not show a difference in the overall amount of the RPul2 protein in the two samples (Fig 7B). One likely explanation is that the amount of free MtRPuL2A protein, which has signal activity, is low in comparison with that in ribosomes. Nevertheless, these results strongly suggest a link between AOI and nodule development.

### Discussion

Here we report evidence that a *Medicago* ribosomal RPuL2 protein, MtRPuL2A, triggers *S. meliloti* cAMP signaling *ex planta*, as does signal 1 in nodules. Sensing of MtRPuL2A by reporter bacteria has the same genetic requirements as signal 1 sensing, thus making it unlikely that activation by MtRPuL2A results from molecular mimicry. Indeed, whereas the *S. meliloti* NsrA receptor protein is involved in recognition of two different signals in symbiosis [17], signal transduction is very specific: signal 1 transduction in nodules specifically requires the CyaK adenylate cyclase whereas signal 1' transduction requires CyaD1 and/or CyaD2 [16, 17]. The fact that MtRPuL2A signal transduction *ex planta* requires CyaK argues for MtRPuL2A being a *bona fide* signal 1. Noteworthy, we have excluded the artifactual dissociation of ribosomes during crude extract preparations and during bioassays as the source of signal activity. We, however, acknowledge the need for *in planta* evidence to ascertain that MtRPuL2A is indeed signal 1, including the generation of down- and up-regulated expression mutants in the corresponding gene. *Mtnf-ya1* nodules are promising material in this respect since we have shown that they essentially lack signal 1 activity. Specific assays are now required to measure free MtRPuL2A/MtRPuL2B protein levels in this material.

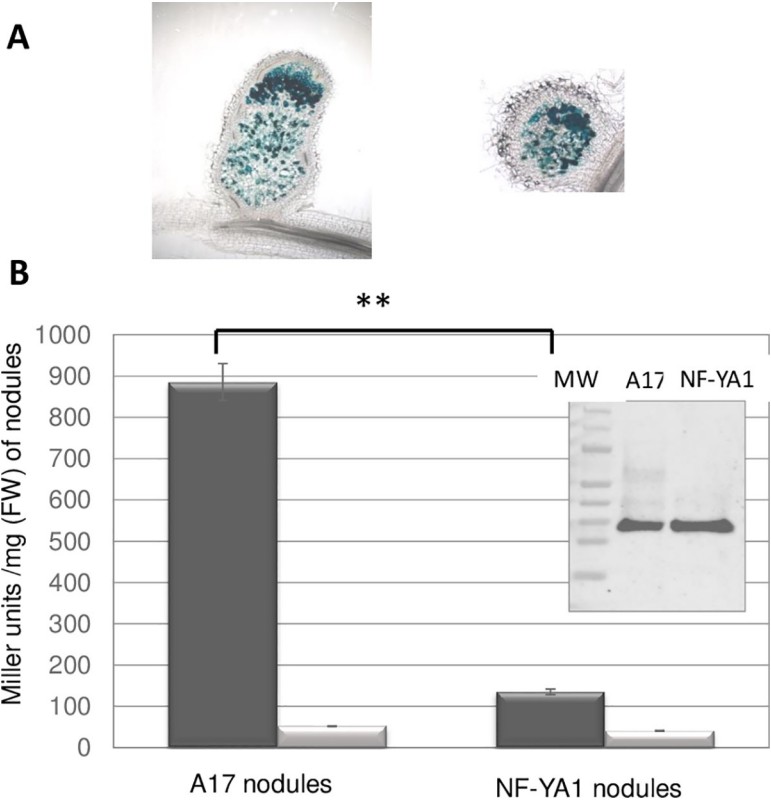

**Fig 7. A *Medicago truncatula Mtnf-ya1* null mutant nodules lack signal 1. Panel A:** A17 (left) and *Mtnf-ya1* (right) nodule sections colonized by a *S. meliloti* wt strain expressing a constitutive *hemA-lacZ* fusion. Endosymbiotic bacteria show a blue coloration. **Panel B:** signal activity of 25mg (FW) of wt (A17) nodules and *Mtnf-ya1* nodules. Reporter strains are GMI12052 (wt, black) or GMI12071 (*cyaK*,(grey). Plants were grown under aeroponic conditions. P-value 0.0020, t-test, n = 3. The insert shows a Western blot of *M. truncatula* A17 and *Mtnf-ya1* nodules with a human anti-RPL8 (RPuL2) antibody.

Another line of future research is the localization of free MtRPuL2A protein in symbiosomes as well as the elucidation of the mechanism by which it would reach the bacteroids *in planta*. Noteworthy, signaling takes place in very young (7dpi) and in the nitrogen-fixing zone (ZIII) of 14dpi nodules [16], thus making it unlikely that signaling takes place during nodule senescence. The lack of a detectable signal peptide in MtRPuL2A (or any other protein of the family) makes the secretion by the nodule-specific secretion system [37, 38] unlikely and suggests secretion by the unconventional secretion pathway [39].

Our data indicate that the MtRPuL2A protein is active in signaling in a non-ribosomal form (Figs 5 and 6), as reported for other moonlighting RPs [40, 41]. Furthermore, it was shown before that free *E. coli* RPul2 is in an unfolded form under physiological conditions [20]. Accordingly, we found that the amino- and carboxy-terminal moieties of *E. coli* RPuL2 both displayed high signalactivity (S3 Fig). We also noticed the high abundance of Hsp 90, a known chaperone of free unfolded RPuL2 in *E. coli*, in the *Medicago* nodule (S2 Table). Altogether, these observations suggest that the free MtRPuL2A signal protein is unfolded in nodules.

Free RPs can be post-translationally modified (*eg* phosphorylated) or complexed with cytosolic proteins, which may affect their activity and turn-over [40, 42]. Since the RNase-A-induced dissociation of purified *Medicago* ribosomes markedly increased signal activity *in vitro*, a post-translational modification of MtRPuL2A is probably not required for its signal

activity. In contrast, the interaction of free MtRPuL2A protein with other proteins *in vivo* may control its turnover or signal activity. Noteworthy, we have observed that the RNAse-A treatment of nodule crude extracts did not significantly increase signal activity (S6 Fig), in contrast to the RNAse treatment of purified ribosomes (Fig 6). We speculate that protein(s) present in the nodule extract may associate with ribosome-liberated MtRPuL2A protein and control its signal activity.

In mammals, 14 different free RPs sequester the ubiquitin ligase MDM2, thereby controlling the fate of the central regulator p53 protein [43]. We do not exclude that other RPs may contribute to signal 1 activity in addition to MtRPuL2A. First, it is possible, if not likely, that the MtRPuL2B isoform (97% amino acid identity with MtRPuL2A) contributes to signal 1 activity since both genes have similar expression patterns (S5 Fig). Second, fiberglass binding of nodule crude extracts depleted only 60% of signal 1 activity (S4 Fig), which may suggest the presence of non-uL2 signal molecule(s) in the nodule extract. Alternatively, it is also possible that the fiberglass depletion assay did not work quantitatively in the complex, nucleic acid-rich, nodule crude extract. More experiments are needed to clarify this point.

The 14 mammalian RPs, which are unrelated in sequence and 3D structure, recognize MDM2 by establishing an electrostatic interaction with its central acidic domain (CAD) [43]. We suggest that MtRPuL2A and NsrA may also interact electrostatically. Both MtRPuL2A and *E. coli* RPuL2 proteins carry a large number of basic residues distributed all along the (unfolded) protein whereas the external loops of the beta-barrel portion of NsrA (the only portion of the protein exposed to the bacterial surface) carry a large number (57) of acidic residues (S7 Fig). The fact that not all basic proteins or compounds can act as signals (Fig 5), however, indicates a specificity in recognition.

In mammals, the accumulation of free RPs in the cytoplasm is the hallmark of ribosome (nucleolar) stress, a cellular response to an alteration in the structure of the nucleolus or in ribosome function/assembly. Ribosome stress is induced by either exogenous (*eg* drugs) or endogenous (*eg* hypoxia, starvation. . .) cues [40, 44]. Evidence for ribosome (nucleolar) stress in plants is only at its beginnings [45] and, to our knowledge, no ribosome stress has been reported so far in the context of symbiosis. We have detected over the years signal 1 activity in nodules of *Medicago* plants grown under a variety of conditions (this study, [16, 17, 46]). Signal 1 thus likely relates to an endogenous, physiological, feature of nodules. The quasi absence of signal activity in *Mtnf-ya1* nodules contrasted with our previous observations that *M. sativa* nodules elicited by an *exoY* mutant of *S. meliloti*, which are small, uninfected, Fix⁻ and senesce early, contained full signal 1 activity [16]. The results obtained with the *Mtnf-ya1* mutant thus strongly suggest a link between AOI and the developmental stage of the nodule, independently of the level of rhizobial infection. The fact that *Mtnf-ya1* roots–but not *exoY*- nodulated *M. truncatula* roots [47]- are hyper-infected [35] also support this conclusion. Many changes occur during nodule development including the establishment of nodule hypoxia [9] and profound alterations in the cell cycle [48] which may result in a ribosome stress, possibly triggering the dissociation of ribosomes and the export of free RPs to the cytosol.

Several RPs have been shown to exert so-called "moonlighting" functions in the cytosol [41]. RP moonlighting is well established in bacteria and animals and, to a lesser extent, in plants (reviewed by [43, 49, 50]. *E. coli* RPuL2 itself has two moonlighting functions in *E. coli*. First, it interacts with the alpha-subunit of RNA polymerase to enhance transcription from the *rrnD* promoter [51]. Second, RPuL2 inhibits chromosome replication *in vitro* upon interacting with DnaA [52]. Free RPL22 contributes to zebra fish and mouse embryogenesis by controlling, together with its paralog RPL22l1, the splicing of pre-mRNA molecules essential for development [53]. Interestingly, a RPL22 homologue (Rpf84) was shown recently to control infection and nodule development in the tree legume *Robinia pseudoacacia*, by an unknown

mechanism [54]. RPL10 also plays a moonlighting function in *Arabidopsis thaliana*. Upon phosphorylation by the LRR-Kinase NIK1, RPL10 suppresses host translation as part of an antiviral immunity mechanism [42].

In conclusion, the present work suggests that the MtRPuL2A protein has a moonlighting signaling function in symbiosis, conceivably connecting the developmental stage of the nodule with the down regulation of root susceptibility to infection events. The fact that most RPs are encoded by 2 to 7 genes in plants, many of which are developmentally regulated [55, 56] calls for further exploration of the role of non-ribosomal RPs in plant biology.

## Materials and methods

### Recombinant plasmids construction

Strains, plasmids and oligonucleotide primers used in this work are described in S4 and S5 Tables, respectively. The pCDFDuet-1 His$_6$-TopA recombinant plasmid was constructed by PCR-amplification of the coding sequence of the *E.coli topA* gene with primers topA/ TOPRIM-F and topA/ZF+ribb-R (S5 Table). The resulting amplicon was cloned at the EcoRV site of pBlueScript II SK(+), digested with SacI and HindIII and ligated into pCDFDuet-1 plasmid digested with SacI and HindIII.

For the construction of pCDFDuet-1 RPuL2-Strep-tag, the coding sequence of *E. coli rplB* was amplified by PCR using the L2BglII-Fw and L2-StrepTAG-rev primers (S5 Table). The resulting DNA fragment was subcloned into the EcoRV restriction site of pBlueScript II SK(+), then digested with BglII and XhoI and ligated into pCDFDuet-1 digested with BglII and XhoI.

For the expression of the amino terminus (residues 1–121) and carboxy-terminal (residues 122–273) fragments of *E. coli* RPuL2, the corresponding coding sequences were PCR-amplified using primers L2BglII-Fw, RevL2-1to121StrepTag and L2-122-BglII-Fw, L2-StrepTAG-rev, respectively. The amplicons were subcloned into the EcoRV restriction site of pBlueScript II SK(+), then digested with BglII and XhoI and ligated into pCDFDuet-1 digested with BglII and XhoI.

The pET22b-*smc02178*-strep-tag recombinant plasmid was constructed by PCR- amplification of the coding sequence of the *smc02178* gene using the *S. meliloti* Rm1021 genomic DNA as a template and the primer pair NdeI 2178 Stp and XhoI 2178 Stp. The PCR product was digested with NdeI and XhoI, purified and ligated into NdeI- and XhoI- digested plasmid pET22b(+).

The pCDFDuet-1-MtRPuL2A-Strep-tag and pCDFDuet-1-MtrunA17Chr8g0347691--Strep-tag plasmid constructs were purchased from GenScript Biotech (Netherlands). Briefly, codon-optimized versions of the MtRPuL2A and MtrunA17Chr8g0347691 coding sequences (but the stop codon) followed by a 30 bp sequence coding for a 2-amino acids linker (AS), the strep-tag (WSHPQFEK) and a stop codon, were cloned at the NdeI and XhoI restriction sites of pCDFDuet-1. All plasmids were verified by Sanger sequencing.

The *S. meliloti* GMI12072 was constructed by introduction of the plasmid pGD2178 into a *nsrA* mutant strain (GMI12049, [17] by triparental mating using pRK600 as a helper plasmid. The *S. meliloti* GMI12071 was constructed by elimination of the gentamycin resistance marker of *cyaK* in GMI11556 [16] by marker exchange using the *sacB* selection procedure [57]. Next, plasmids pGD2178 and pGMI50333 were introduced by triparental mating using pRK600 as a helper plasmid. Strains genotype is described in S4 Table.

### Plant material and culture conditions

Seeds of *M. truncatula* Jemalong A17 and *Mtnf-ya1.1* mutant [35] and *M. sativa* cv Gemini NAR [58], were scarified by immersion in concentrated H$_2$SO$_4$ during 5–7 min, washed 3

times with sterile water, surface-sterilized with a diluted (1:4) commercial bleach solution for 2 min and thoroughly washed again with sterile water. Seeds were then placed in 0.8% (w/v) water-agar plates and kept 3 days at 4˚C for synchronization of germination. Plantlets were grown in aeroponics and inoculated with $2 \times 10^5$ *S. meliloti* cells/ml. *M. sativa* NAR plants were grown in sterile pots containing sepiolite. Nodules from *M. truncatula* were harvested 21–30 days after inoculation and 15 days for *M. sativa*. Nodules were immediately frozen in liquid nitrogen and stored at -80˚C. Nodules sections (Fig 7) were prepared as described before [17].

## Bioassay for signal activity

We monitored signal activity by quantifying expression of the *smc2178*::*lacZ* reporter fusion in a *S. meliloti* strain overexpressing the *nsrA* receptor protein (GMI12052) or an isogenic *cyaK* mutant derivative (S4 Table). A bacterial suspension ($OD_{600}$ = 0.1) was obtained by diluting an overnight culture of the reporter strains in synthetic modified Vincent medium [59] supplemented with gentamicin (20µg/ml) and tetracycline (10µg/ml). 950 µl of this suspension was mixed with 50 µl of the signal solution to be tested in a sterile polystyrene tube, incubated overnight at 28˚C in a rotatory shaker (200 rpm). NCR peptides were assayed at the highest concentration (0.8 µM each) that did not impair bacterial growth, Histones at 0.33 µM and polymyxin B sulfate (Sigma Aldrich) at 0.36 µM. β-galactosidase activity was quantified as described before [16]. For specific activities assessment, the protein content of the assayed sample was quantified by the Bradford method (Bio-rad, USA). We adopted the following rule to illustrate statistical significance in figures: *, $P < 0.05$; **, $P < 0.01$; ***, $P < 0.001$; actual P-values are given in the text or in figure legends.

## Signal extraction from microbes

*E. coli* (DH5α) was grown at 37˚C in LB medium. *S. meliloti* (Rm1021) and *S. cerevisiae* (BY4741) were grown at 28˚C in LB medium supplemented with 2.5 mM $CaCl_2$ and 2.5 mM $MgSO_4$ and YPD medium, respectively. Bacterial cell pellets from 50 ml overnight cultures were washed with lysis buffer (20 mM $Na_2PO_4$/$NaH_2PO_4$ pH 8; 100 mM NaCl; 1 mM DTT; 0.5 mM EDTA; 1 mM PMSF) and suspended in 20 ml of the same buffer. Bacterial cells kept on ice were broken by sonication (Branson Sonifier 250 equipped with a macro-tip, 10 s cycles x 10 times, Power 6) until the suspension became clear. Washed yeast cell pellets kept in safe-lock tubes with 3 metal beads were frozen in liquid nitrogen and cryogenically grinded (2 x 60 s, 30 cycles.s$^{-1}$) in a Mixer Mill MM 400 (Retsch,Germany). The resulting powder was suspended in lysis buffer. All lysates were clarified by centrifugation (15000 g, 4˚C, 30 min) and sterilized by filtering through a 0.22 µm pore membrane (Millipore, Germany). For protease sensitivity assays, 1 mg of Immobilized proteinase K (Sigma-Aldrich) was incubated with 100 µl of crude extract at 37˚C. After a brief spin, the supernatant was aspirated and tested for signal activity.

## Signal purification from *E. coli* extracts

A clarified bacterial crude extract was prepared from 1 l of *E. coli* DH5α (S4 Table), as described above. All purification steps were carried out by FPLC (Äkta purifier-10, GE Healthcare). The clarified crude extract was loaded onto a 5 ml hydroxylapatite column (mini CHT type I, Bio-Rad) preequilibrated with buffer A (20 mM $Na_2PO_4$/$NaH_2PO_4$, 100 mM NaCl, pH 8). Signal activity was step-eluted with NaCl (400 mM step). Active fractions were pooled (*ca.* 25 ml) and diluted with 50 ml of buffer A before loading onto a Hi-Trap Q HP 1 ml column, the flow-through of which was loaded on a Hi-Trap heparin HP 1 ml column (GE Healthcare)

pre-equilibrated with buffer B (20 mM $Na_2PO_4/NaH_2PO_4$, 200 mM NaCl, pH 8). Signal activity was eluted from the heparine column with a linear NaCl gradient (0.2 to 2M). Active fractions were pooled and further separated by gel filtration on a Superdex 75 16/600 preparative grade gel filtration column (GE Healthcare) pre-equilibrated with buffer A. Fractions showing signal activity were concentrated on a Hi-Trap SP HP 1 ml column (GE Healthcare) column, previously equilibrated with buffer A. Signal activity was eluted by a linear NaCl gradient (0.1 to 1 M). Protein fractions were further analyzed on SDS-PAGE stained by InstantBlue[TM] Commassie (Expedeon).

## Signal extraction from root nodules

50 mg of frozen nodules were crushed with mortar and pestle and extracted with 500 μl of lysis buffer (20 mM $Na_2PO_4/NaH_2PO_4$, 100 mM NaCl, 1 mM DTT, 0.5 mM EDTA, 1 mM PMSF, pH 8). The extract was centrifuged at 15000 g during 10 min at 4°C, the pellet was extracted again as described before and the supernatants were pooled, filtered through a 0.22 μm membrane (Millipore) and kept on ice.

## Purification of $His_6$-TopA

The pCDFDuet-1 $His_6$-TopA was introduced into *E. coli* BL21-Rosetta (DE3)-pLysS ($Cm^R$) cells (Novagen) (S4 Table) by electroporation and plated on LB medium supplemented with glucose 0.2%, chloramphenicol (12.5 μg/ml) and streptomycin (50μg/ml). The resulting strain was grown at 37°C in LB supplemented with streptomycin (50μg/ml) and chloramphenicol (12.5 μg/ml) at 37°C until the $OD_{600}$ reached 0.7. Recombinant protein expression was induced with 1 mM IPTG followed by growth at 37°C for 1 h. Pellets were collected by centrifugation before liquid nitrogen freezing. Clarified crude extracts were prepared as described before in lysis buffer (20 mM $Na_2PO_4/NaH_2PO_4$ pH 8; 10 mM imidazole, 500 mM NaCl; 1 mM DTT; 0.5 mM EDTA; 1 mM PMSF). Protein purification was performed by FPLC (Äkta purifier-10, GE Healthcare) with a HisTrap Nickel HP 1 ml column (GE Healthcare) pre-equilibrated with buffer containing 10 mM imidazole, then washed with 20 mM imidazole and eluted with a linear gradient of 20 mM to 500 mM imidazole. Enriched fractions were diluted 1:5 with buffer A and loaded into a Hi-Trap heparin HP 1 ml column (GE Healthcare) pre-equilibrated with the same buffer. After washing with the same buffer, the recombinant protein was eluted with a linear NaCl gradient (0.1M to 0.5 M).

## Expression and purification of Strep-tagged proteins

In all cases, recombinant plasmids were introduced in *E. coli* BL21-Rosetta (DE3)-pLysS ($Cm^R$) cells (Novagen) by electroporation and plated in LB supplemented with glucose 0.2%, chloramphenicol (12.5 μg/ml) and streptomycin (50μg/ml). 5–10 colonies were pooled and used to inoculate 200 ml of the same medium at 37°C, or 16°c for MtRPuL2A-Strep-tag and MtrunA17Chr8g0347691-Strep-tag, on a rotatory shaker at 200 rpm until an $OD_{600}$ = 0.7. For expression of *E. coli* RPuL2-Strep-tag or Smc2178- Strep-tag the culture was centrifuged and the pellet was resuspended in LB supplemented with streptomycin (50μg/ml) and IPTG (1 mM) and further grown during 1h. For MtRPuL2A-Streptag and MtrunA17Chr8g0347691-Strep-tag, the cultures were supplemented with 0.5 mM IPTG and further grown at 16°C for 4 h.

Cell pellets were resuspended in 20 ml of 50 mM Tris-HCl pH 8, 500 mM NaCl, 1 mM DTT, 1 mM EDTA, 1 mM PMSF. Clarified crude extracts (prepared as described above) were loaded into a pre-equilibrated Strep-Tactin®column (column volume 200 μl). Bound protein was washed with the same buffer and eluted with 5 mM desthiobiotin. *E. coli* RPuL2-Strep-tag

was loaded on a buffer exchange column (Healthcare) and eluted with a buffer containing 50 mM Tris-HCl pH 8, 100 mM NaCl. Further purification was achieved by FPLC (Äkta purifier-10, GE Healthcare) on a Hi-Trap heparin HP 1 ml column (GE Healthcare). Bound RPuL2--Strep-tag was washed with buffer and eluted with a linear NaCl gradient (0.2-1M). RPuL2-enriched fractions were pooled and further purified by gel filtration on a S75 column (GE Healthcare) pre-equilibrated with a 50 mM Tris-HCl pH 8, 100 mM NaC buffer.

## Ribosome purification

Clarified crude extracts of *Medicago* nodule (100 mg FW) and *E.coli* cells were prepared as described above, except that the lysis buffer contained 10 mM Tris-HCl pH 7.4, 70 mM KCl, 10 mM MgCl$_2$. A chromatographic method for the isolation of ribosomes based on the use of strong anion exchange (QA) monolithic columns (BIA Separations, Slovenia) was used according to [31].

## Fiberglass binding assays

Samples were loaded on a column containing 40 mg of fiberglass pre-equilibrated with the appropriate buffer. The flow-through was collected by gravity and tested for signal activity, SDS-PAGE and Western blot analyses. Column-bound material was washed 3 times before elution with 1X Laemmli loading buffer for SDS-PAGE and Western blot analyses.

## Western blot analyses

Samples were mixed with 4x Laemmli buffer, denatured 3 min at 95˚C and loaded on SDS-PAGE precast 4–15% gels (Mini-PROTEAN TGX gel, Bio-Rad) followed by electrophoresis at 200 V for 30 min in 1X TGS (Tris 2.5 mM, Glycine 19.2 mM, 0.01%SDS, pH8.3) buffer. After migration, proteins were electrotransferred to a nitrocellulose membrane (Amersham Protran 0.45 µm; GE Healthcare) during 1 h at 20 mA. The membranes were probed with rabbit Anti-RPL8 IgG antibodies (HPA050165; Sigma Prestige Antibodies, 1:1000) and then incubated with a secondary anti-rabbit IgG coupled to HRP (1:10,000). Alternatively, membranes were incubated with a Strep-Tactin-HRP conjugate (IBA Life Sciences, 2-1502-001, 1:2000). Membranes were incubated with a chemiluminescence substrate (Bio-Rad) and imaged on a ChemiDoc MP imager (Bio-Rad).

## Mass spectrometry analyses

**Digestion and nano-LC-MS/MS analysis.** Samples were reduced using Laemmli buffer supplemented with 30 mM DTT at 56˚C for 30 min. Cysteines were alkylated by the addition of 90 mM iodoacetamide for 30 min at room temperature. Protein samples were loaded onto a 12% SDS-polyacrylamide gel and subjected to short electrophoresis (~0.5 cm). After Instant Blue® (Invitrogen) staining of the gel, gel bands were excised, washed twice with 50 mM ammonium bicarbonate-acetonitrile (1:1 v:v) and washed once with acetonitrile. Proteins were in-gel digested at 37˚C overnight by the addition of 60 µl of a solution of modified sequencing grade trypsin in 25 mM ammonium bicarbonate (10 ng/µl, sequence grade, Promega, France). The resulting peptides were extracted from the gel by one round of incubation (15 min, 37˚C) in 1% formic acid–40% acetonitrile and two rounds of incubation (15 min each, 37˚C) in 1% formic acid–acetonitrile (1:1). The three extracted fractions were pooled and air-dried. Tryptic peptides were resuspended in 14 µl of 2% acetonitrile and 0.05% trifluoroacetic acid (TFA) for MS analysis.

The peptide mixtures were analyzed by nano-LC-MS/MS using a nanoRS UHPLC system (Dionex, Amsterdam, The Netherlands) coupled to an LTQ-Orbitrap Velos mass spectrometer (Thermo Fisher Scientific, Bremen, Germany). 5 µl of each sample were loaded on a C18 precolumn (300 µm inner diameter x5 mm, Dionex) at 20 µl/min in 5% acetonitrile, 0.05% tri-fluoroacetic acid. After 5 min desalting, the precolumn was switched on line with the analytical C18 column (75 µm inner diameter x 15 cm; in-house packed) equilibrated in 95% solvent A (5% acetonitrile, 0.2% formic acid) and 5% solvent B (80% acetonitrile, 0.2% formic acid). Peptides were eluted using a 5 to 50% gradient of solvent B over 105 min at a 300 nl/min flow rate. The LTQ-Orbitrap was operated in data-dependent acquisition mode with the Xcalibur software. Survey scan MS spectra were acquired in the Orbitrap on the 300–2000 m/z range with the resolution set to a value of 60,000. The 20 most intense ions per survey scan were selected for CID fragmentation, and the resulting fragments were analyzed in the linear trap (LTQ). Dynamic exclusion was used within 60 s to prevent repetitive selection of the same peptide.

**Database search and data analysis.** The Mascot Daemon software (version 2.6.1, Matrix Science, London, UK) was used to perform database searches in batch mode with all the raw files acquired on each sample. To automatically extract peak lists from Xcalibur raw files, the Extract_msn.exe macro provided with Xcalibur (version 2.2 SP1.48, Thermo Fisher Scientific) was used through the Mascot Daemon interface. The following parameters were set for creation of the peak lists: parent ions in the mass range 400–4500, no grouping of MS/MS scans, and threshold at 1000. A peak list was created for each analyzed fraction (*i.e.* gel slice), and individual Mascot searches were performed for each fraction. Data were searched against all entries from the *E.Coli* UniProtKB database (Swiss-Prot/TrEmbl release 20150320, 547964 entries) or against a custom Medicago truncatula database (release 20170704, 44623 entries). Oxidation of methionine and carbamidomethylation of cysteine were set as variable modifications for all Mascot searches. Specificity of trypsin digestion was set for cleavage after Lys or Arg except before Pro, and one missed trypsin cleavage site was allowed. The mass tolerances in MS and MS/MS were set to 6 ppm and 0.8 Da, respectively. The instrument setting was specified as "ESI-Trap." Mascot results were parsed and validated with a in-house developed software called Proline (version 1.6) available at http://proline.profiproteomics.fr/.

The target-decoy database search allowed us to control and estimate the false positive identification rate of our study, and the final catalogue of proteins presented an estimated false discovery rate (FDR) below 1% for peptides and proteins. To increase robustness, only proteins identified with at least 2 peptides and 4 MS/MS were considered as correctly identified.

## Phylogenetic analyses

RPuL2 homologs were found from NCBI blasts and examination of the latest release of the *M. truncatula* genome [60]. This latter analysis revealed that the annotation of 6 *M. truncatula* RPuL2 proteins (MtrunA17CPg0492511.2, MtrunA17Chr3g0090931.2, MtrunA17Chr3g0094621.2, MtrunA17Chr4g0016021.2, MtrunA17Chr4g0017201.2, MtrunA17Chr4g0024461.2) needed adjusting by joining pairs of protein-coding regions each time via Group II introns. These sequences have been deposited in GenBank under the accession number (XXXX). The tree in Fig 3 was computed using https://ngphylogeny.fr (PhyML+SMS) [61], and visualized with Itol [62].

## Supporting information

**S1 Fig. Identification of RPuL2 as *E. coli* signal 1. Panel A**: Signal 1 purification from *E. coli* DH5a crude extracts. From top to bottom: Purification chart flow, activity assay and SDS-PAGE analysis of fractions eluted from the SP column (B6 corresponds to the 0.3M NaCl fraction). **Panel B**: Signal 1 purification from a *E. coli* strain overexpressing a TopA-his tagged

protein. From top to bottom: purification flowchart, activity and SDS-PAGE analysis of fractions eluted from the heparin column (G12 corresponds to the 0.32M salt fraction). See methods for details. The black and grey arrowheads points to the RPuL2 and the TopA-His proteins, respectively. The black arrowhead band was excised and identified by MS analysis as being RPuL2 (RplB) (see S1 Table).
(PPTX)

**S2 Fig. Signal activity in control experiments. Panel A**: Synthetic Strep-Tag® peptide (SAWSHPQFEK; 60 µg) activity. B control buffer. **Panel B**: Activity of a purified SMc02178-Strep-Tag® protein compared to RPuL2 activity. 1µg of each protein was assayed. **Panel C**: Activity and SDS-PAGE analysis of protein fractions in a mock purification (empty vector) assay on a Strep-Tactin® resin. B control buffer. CE crude extract. MW molecular weight ladder. F flow-through of the Strep-Tactin® resin. W last wash of the Strep-Tactin® resin. E1-E6 elution fractions.
(PPTX)

**S3 Fig. Amino- and carboxy-terminal moieties of *E. coli* RPuL2 display signal activity.** Specific activities of purified amino-terminal (1–121) and carboxy-terminal (122–273) moieties of *E. coli* RPuL2 (10-fold dilution of the main elution fraction from the Strep-Tactin® column. Both proteins were strep-tagged at the carboxy-terminal end.
(PPTX)

**S4 Fig. Fiberglass signal depletion on *Medicago sativa* nodule extracts.** Signal activity of the Input (I) and flow-through (F) fractions of a fiberglass column. B buffer control. P-value 0.0078, *t*-test, n = 5. The right panel features a representative western blot using anti-human RPL8 protein antibodies. Please note that the human anti-RPL8 antibodies cannot detect low amounts of heterologous RPuL2 proteins.
(PPTX)

**S5 Fig. Expression analysis of *MtRPuL2A* (*MtrunA17Chr7g0247311*, red) and *MtRPuL2B* (*MtrunA17Chr5g0405281*, blue). Panel A**: Affymetrix normalised hybridisation data for a selection of conditions (nodules, leaves, shoots, roots. . .), extracted from GeneAtlas (https://mtgea.noble.org/v3) showing the strong correlation (0.86) in expression profiles of the two probes. **Panel B**: Total polyA reads in RNAseq data from 10 day old nodules or roots (from [27]. **Panel C**: RNAseq counts of ribo-depleted RNA samples from laser microdissected nodule zones (from [27].
(PDF)

**S6 Fig. RNAse and RNAse inhibitors treatments do not impact nodule extracts signal activity. Panel A:** RNAse treatment of Mt A17 and *Mtnf-ya1* nodule extracts n = 3. **Panel B**: Effect of RNASe inhibitors (P-value = 0.046, t-test, n = 3). 25 mg of nodules fresh weight were used per assay.
(PPTX)

**S7 Fig. Postulated mode of recognition between MtRPuL2A and *S. meliloti* NsrA. Panel A:** Charge distribution of selected ribosomal proteins according to Pepcal (https://pepcalc.com). Color code for amino acids: red acidic, green aromatic, cyan basic, dark green polar. Top line is hydrophilic, bottom line is hydrophobic. **Panel B** visualisation of acidic residues (DE, underlined yellow) in the surface exposed (blue) and inner loops (green) of the beta-barrel portion of NsrA (amino acids 600–1200). TM regions are shown red. Loop prediction was done using the Pred-TMBB software.
(PPTX)

**S1 Raw images.**
(PDF)

**S1 Table. Mass spectrometry analysis of the signal active fraction in *E. coli* crude extracts.**
(XLSX)

**S2 Table. Mass spectrometry analysis of *Medicago* A17-*S. meliloti* 1021 nodule extracts after fiberglass binding.**
(XLSX)

**S3 Table. Mass spectrometry analysis of purified *Medicago sativa* NAR ribosomes.**
(XLSX)

**S4 Table. Bacterial strains and plasmids used in this study.**
(DOCX)

**S5 Table. Oligonucleotide primers used in this study.**
(DOCX)

## Acknowledgments

We are grateful to the following colleagues for their help all along this work: Dr A. Henras and Dr C. Plisson (LBME Toulouse) for sharing their expertise on ribosomal proteins, Dr D. Trouche (LBME Toulouse) for the gift of histone proteins, Dr P. Mergaert (I2BC Gif sur Yvette) for the gift of purified NCR peptides, Dr A Niebel (LIPM) for the gift of *Mtnf-ya1.1* seeds and for discussions, Dr P. Gamas and J. Gouzy (LIPM) for early access to the *M. truncatula* v5 genome database, E. Sallet and S. Carrrère (LIPM) for reannotation of 6 *M. truncatula* protein-coding sequences, Dr C. Masson (LIPM Toulouse) and Dr P. Batut (Princeton Univ.) for useful comments on the manuscript.

## Author Contributions

**Conceptualization:** Jacques Batut.

**Data curation:** Fernando Sorroche, Jacques Batut.

**Formal analysis:** Fernando Sorroche, Patrice Polard, Jacques Batut.

**Funding acquisition:** Clare Gough, Jacques Batut.

**Investigation:** Fernando Sorroche, Violette Morales, Saïda Mouffok, Carole Pichereaux, A. Marie Garnerone, Lan Zou, Badrish Soni, Marie-Anne Carpéné, Audrey Gargaros, Fabienne Maillet, Odile Burlet-Schiltz, Verena Poinsot, Clare Gough.

**Methodology:** Violette Morales, Carole Pichereaux, Verena Poinsot, Clare Gough, Jacques Batut.

**Project administration:** Jacques Batut.

**Resources:** Violette Morales, Carole Pichereaux, Verena Poinsot, Patrice Polard.

**Supervision:** A. Marie Garnerone, Patrice Polard, Clare Gough, Jacques Batut.

**Validation:** Fernando Sorroche, Clare Gough, Jacques Batut.

**Writing – original draft:** Jacques Batut.

**Writing – review & editing:** Fernando Sorroche, Violette Morales, Patrice Polard, Clare Gough, Jacques Batut.

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
