## [Decision Letter · Decision Letter 0]

1 Apr 2020

PONE-D-20-05716

The ex planta signaling activity of a Medicago ribosomal uL2 protein suggests a moonlighting role in controlling secondary rhizobial infection

PLOS ONE

Dear %Dr% %Batut%,

Thank you for submitting your manuscript to PLOS ONE. After careful consideration, we feel that it has merit but does not fully meet PLOS ONE’s publication criteria as it currently stands. Therefore, we invite you to submit a revised version of the manuscript that addresses the points raised during the review process.

We would appreciate receiving your revised manuscript by May 16 2020 11:59PM. To enhance the reproducibility of your results, we recommend that if applicable you deposit your laboratory protocols in protocols.io, where a protocol can be assigned its own identifier (DOI) such that it can be cited independently in the future. For instructions see: http://journals.plos.org/plosone/s/submission-guidelines#loc-laboratory-protocols

We look forward to receiving your revised manuscript.

Kind regards,

Francisco Martinez-Abarca, Ph.D.

Academic Editor

PLOS ONE

Journal Requirements:

2.

PLOS ONE now requires that authors provide the original uncropped and unadjusted images underlying all blot or gel results reported in a submission’s figures or Supporting Information files. This policy and the journal’s other requirements for blot/gel reporting and figure preparation are described in detail at https://journals.plos.org/plosone/s/figures#loc-blot-and-gel-reporting-requirements and https://journals.plos.org/plosone/s/figures#loc-preparing-figures-from-image-files. When you submit your revised manuscript, please ensure that your figures adhere fully to these guidelines and provide the original underlying images for all blot or gel data reported in your submission. See the following link for instructions on providing the original image data: https://journals.plos.org/plosone/s/figures#loc-original-images-for-blots-and-gels.

Reviewers' comments:

Reviewer's Responses to Questions

**Comments to the Author**

1. Is the manuscript technically sound, and do the data support the conclusions?

Reviewer #1: Yes

Reviewer #2: Partly

2. Has the statistical analysis been performed appropriately and rigorously? 

Reviewer #1: Yes

Reviewer #2: I Don't Know

3. Have the authors made all data underlying the findings in their manuscript fully available?

Reviewer #1: Yes

Reviewer #2: Yes

4. Is the manuscript presented in an intelligible fashion and written in standard English?

Reviewer #1: Yes

Reviewer #2: Yes

5. Review Comments to the Author

Reviewer #1: This work is a follow-up of previous investigations performed by the group focused on the autoregulation of infection (AOI) in the Medicago-Sinorhizobium meliloti symbiosis. The manuscript reports on the identification of one of the two plant signals (specifically signal 1) involved in the AOI in Medicago. By using biochemical approaches and a previously established bioassay, the authors identify a conserved Medicago ribosomal protein, which in a non-ribosomal form exhibits ex-planta signaling activity with the same requirements as nodule signal 1. In addition, the authors find evidence for a link between root susceptibility to rhizobial infection and nodule development based on the lack of signal 1 activity in a Medicago mutant, which shows a hyperinfection phenotype.

This is an interesting study that advances our knowledge about the components that participate in the regulatory circuits that control infection of legumes by rhizobia. The manuscript is well written, experiments have been conducted with rigor and the conclusions are supported by the results. I only have minor comments.

1. Several times in the manuscript, the authors indicate that signal activities are expressed as Miller units per µg of protein. For clarity, this should also be indicated in the Y axis of the different graphs showing signal activities (e.g. Fig. 1, Fig. 2B, Fig. 4, Supplementary Figures…).

2. Line 142: From the 29 remaining candidate proteins from E. coli, for which there were no available mutants in the Keio collection, the authors decided to test the signaling activity of topoisomerase I. Was there any rationale behind this decision? If there was, it should be indicated in the text.

3. Fig. 3B: Since E. coli RPuL2 shows signal 1 activity, it might be informative to see the alignment of the two Mt proteins with that of E. coli RPuL2. Is there any reason for the difference found in solubility/stability and activity between RPuL2 proteins from E. coli and Medicago?

4.- Error bars are missing in most of the graphs. They should be included.

Additional comments:

-Line 75: Please define IRLC

-Lines 249-250: Please rephrase the sentence.

-Line 373. Please correct MDM2

-Fig. 1: Why was not the cyaK mutant included in the assays shown in Fig. 1B?

-Fig. 5: Is the title of the Y axis correct or should it be Miller units/µg?

-Lines corresponding to vertical and horizontal axis need to be drawn in several graphs. Please, revise.

-Italics need to be used for the names of restriction enzymes in Materials and Methods and for species names in the References.

Reviewer #2: Review on Sorroche et al.: The ex planta signaling activity of a Medicago ribosomal ul2 protein suggests a moonlighting role in controlling secondary rhizobial infection

The authors recently described a regulatory circuit that controls secondary infection of Medicago sp. by Sinorhizobium meliloti bacteria. Previous work of the authors showed that this so-called autoregulation of infection includes signals of the host plant that trigger cAMP signaling in bacteroids. As a consequence, the bacteria stimulate the plant’s synthesis of ethylene which is known to inhibit rhizobial infection. However, the plant signals inducing cAMP signaling remained unknown. In this manuscript, the authors found that the ribosomal protein MtRPul2A of Medicago displays such signal activity.

In general, the manuscript is clearly understandable and well written. However, I have some comments/suggestions that may help to improve the quality of this manuscript.

1. Include error bars in Figures 2, 4 and 5 to indicate that the shown data are reproducible. Explain what error bars mean (SD or SE). Provide detailed information on the number of performed experiments and the number of independent samples in each experiment.

2. The fact that compounds from plants, bacteria and fungi induce cAMP signaling in S. meliloti does not necessarily mean that these organisms produce the same ubiquitous signal and that this signal is “widespread”. Also, the title of Fig. 6 “Free RPul2 is the signal” is too strong.

3. The abstract of this paper suggests that the authors have performed localization studies of MtRPul2A and MtRPul2B in nodules. This is somehow misleading because this manuscript does not provide new data (the information has been taken from a previous publication).

4. The beginning of the Result section is difficult to understand. The bioassay (used strains etc.) for signal activity 1 should be better explained at this stage (the Materials and Method section comes after the Result section in this journal). It should also be mentioned in each Figure legend what “signal activity” or “specific activity” (Fig. 5) means (indicate the used test strain etc.). Is it “signal activity”, “signaling activity” or “signal 1 activity”?

5. It is surprising that the signal activity of E. coli RPuL2 did not relate to a specific functional domain of the protein. Are there any repeats in the protein?

6. The file size of Fig. S5 is too big (28 MB). I could not open and review this file.

7. Consider improving the writing of the first part of the Abstract. It’s not clear that the signals 1 and 1’ induce cAMP signaling in bacteroids and that this induces ethylene production and therefore AOI in roots. Avoid too long sentences in the Abstract. “

8. The figures require certain revision (see details below).

Specific comments

40 Delete “identified” and insert “was identified” in the end of the sentence.

42 Explain better what “signaling activity ex planta” means.

49 Delete “and”. Based on …., we suggest…

60 I disagree with this sentence. We know a lot on signals, such as flavonoids, LCOs etc.

76 nodule cysteine-rich

102-103 The reader cannot understand this sentence at this stage. What means “signaling activity ex planta that requires…”

111 Which additional bacteria have been examined?

126 Describe used reporter strains in the text.

127 , grey… delete “here”

133 immobilized-proteinase K… .

183 Provide information on used strains.

222 protein sequences from

249 As RPs are very basic proteins…

262 What means NAR?

277 , thus suggesting that MtRPuL2 is a non-ribosomal protein under…

320 Provide information on the used antibody (origin, specificity)

327 blue coloration

329 were grown under aeroponic…

337 What means “bacterial genetic requirements” in this context?

339 is involved in recognition of two different…

354 It could be discussed here (or later, see line 400) that cytoplasmic (non-ribosomal) MtRPul2A eventually comes into contact with the bacteroids when symbiosome membranes are degraded in senescent nodule cells. The Mtnf-ya1 mutant seems to possess a smaller nodule senescence zone compared to the wild type nodule. In fact, subcellular protein localization studies using fluorescence-tagged MtRPul2A could provide further information on this problem. Measurement of signal activity using tissue derived from different nodule zones could provide additional information in this context.

363 How are RPs post-translationally modified?

383 What means MDM2?

395 Consider deleting “over the year”? In this study? Reported in a previous paper? Add reference.

339 Fix-

408 Replace author name by reference number.

414 Rpf24 or Rpf84? Are there any similarities between this protein and MtRPuL2? Does this protein also belong to the ul2 family? If yes, include this protein into the tree of Fig. 3A.

454-459 Provide better strain description in a way that readers would be able to repeat construction of these strains and refer to Table S4.

479-480 Add spaces between numbers and units.

482 Which figures? Which statistical test?

573 What’s the specificity of this antibody? Which RPLs are expected to cross-react?

635 Include accession number(s).

634 Provide information on used computer program.

666 Please check references again. There are several style errors.

Fig. 1A, 1C 2B, 4B, 4C, 6 Miller units/μg of protein (units in 1C)

Fig. 1B 1h (not 1H)

Fig. 2B Write fraction numbers below columns. Error bars?

Fig. 2C Describe used mutants in figure legend. Error bars?

Fig. 1D Error bars?

Fig. 3 Poor resolution: Increase pixel number

Fig. 3A Would it be possible to include accession numbers? The text explaining the scale bar is too small.

Fig 3B Would it be possible to include E. coli RPul2 into this alignment?

Fig. 4A What means 25 and 35? kDa?

Fig. 4B and 4C: Include error bars.

Fig. 5 Include error bars. Would it be possible to use here also Miller units/μg of protein?

Fig. 7 Miller units/mg (FW) of nodules?

Fig. 7B It would be more informative to show here the whole Western blot. Are there any other bands (proteolytic degradation products)?

------------------

6. PLOS authors have the option to publish the peer review history of their article (what does this mean?). If published, this will include your full peer review and any attached files.

Reviewer #1: No

Reviewer #2: Yes: Christian Staehelin, Sun Yat-sen University, China

---

## [Author Response · Author response to Decision Letter 0]

11 May 2020

Dear Colleague,

On behalf of all authors, I thank you very much for evaluating our work and for the reviewers’ useful criticisms on the manuscript. We have taken all the suggestions and comments into best account and we thus hope that the paper can now be accepted for publication. 

With best wishes,

Jacques Batut

Specific responses to Editor’s comments

1. The manuscript, including the front page and the references section, has been thoroughly edited to conform to PLOS One style requirements. We did not italicize restriction enzymes names and species names in the reference list (see reviewer #1 comment). Of course, we can change this, if needed.

2. Gel and blot Images : 

Fig 2AD, 4AC and 7, S1AB, S2C, S4: the images shown correspond to the original full-length images with no or very minimal cropping. The insert in Fig 5 was a merge of two SDS-PAGE gel images. To clarify this, we have put a vertical line between the two samples on the revised Fig. 5 to and have modified the figure legend as follows “The insert shows SDS-PAGE images of purified protein preparations. All the original images have been compiled in a supplementary pdf file called S8 raw images.

3. L. 151*. We have removed the “data not shown” phrase referring to the preliminary observation that led us to investigate the signal activity of E. coli topoisomerase I. Full data regarding E. coli TopI activity are shown in S1 Fig (panel B). This section was also rephrased to comply with reviewer #1 request (see point 2 below).

* Line numbers refer to the unmarked copy.

Reviewer#1

1. The units are now specified in the Y axis directly. 

Our strategy was to assay the signal activity of as many as possible of the 59 candidate proteins identified in S1 Table. For 30 of them, mutants existed in the Keio collection. For topoisomerase I, no mutant was available (essential gene) but the purified protein existed as a commercial supply (Promega). This part has been rephrased as follows for clarity (l. 146, line numbers s correspond to the marked manuscript). “Instead, we found that a commercial preparation of E. coli topoisomerase I (TopA, Promega) had a significant signaling activity. The corresponding mutant did not exist in the Keio collection, as expected since top1 is an essential gene in E. coli. To validate this observation, we therefore extensively purified an amino-terminal His6-tagged version of E. coli TopA on Nickel and Heparine-sepharose columns (S1 Fig).”

2. E. coli and M.truncatula RPuL2, as orthologous proteins (Fig 3A), display sequence similarity that reflects their phylogenetic relatedness but probably says nothing on the molecular determinants of signal activity. We argue in the paper (l. 162-168 and S3 Fig) that activity does not rely on a specific motif /domain of the RPuL2 proteins. Instead, we propose (l. 396-403 and S7 Fig) that basic residues/patches at the surface of both E. coli and Medicago RPuL2 proteins may account for activity.

The fact that the E. coli and Medicago proteins were (over)produced in a homologous and heterologous genetic context (E.coli), respectively, may account for the difference in activity/solubility eg as a folding or post-translational modification issue. 

3. Fig 2, 4, 5: Error bars have been added to Fig 2C, Fig 4 (a new set of data has been included) and Fig 5 as well as in SI Figures, whenever possible. 

Additional comments:

IRLC defined l. 76

L. 256: Rephrased as suggested by reviewer #2: “As RPs are very basic proteins basic (pI 11.1 for MtRPuL2A/MtRPuL2B vs pI 9.45 for MtrunA17Chr8g0347691), we tested other cationic compounds for signal activity.”

L. 387: MDM2 corrected 

Fig 1B: cyaK-dependency of the Medicago nodule signal was established in Fig 1A. Fig. 1B aims at testing the effect of proteinase K treatments on signal activity.

Fig 5: Y axis: µM is correct. Since we aim to compare the signal activities of peptides/proteins of very different molecular sizes, µg cannot be used.

Y and X axis now drawn in all Figures 

Our understanding is that PLOS One does not use italics for restriction enzymes following Roberts et al. (2003) NAR 31: 1805-1812. 

Similarly, it seems that species names should not be italicized in the reference list in PLOS One. 

Reviewer#2

1. Error bars have been added in Fig 2, 4 and 5. Error bars feature SE, this is now indicated in the figure legends as well as the number of independent assays performed and the statistical test used. 

2. L.111 “Signal 1 is ubiquitous…” We agree that, at this stage of the paper, ubiquity only relies on cyaK-dependency. We changed “ubiquitous” for “widespread” to tune down the assertion. In the Legend of Fig 6 (L. 309) . “Free RPuL2 is the signal” was changed for “Free RPuL2 has signal activity”. 

3. Abstract: We removed the sentence regarding candidate gene expression data as this is not our own experimental data.

4. The rationale for the bioassay is described in the Introduction section with appropriate references (l. 92-97). For clarity, we have now added the following sentence at the beginning of the results section (l. 113-114) “We used a S. meliloti strain (GMI12052, S4 Table) that overproduces the NsrA receptor protein as a reporter strain for signal activity.” The used strains are now specified in all the figure legends and their genotype described in detail in S4 Table. 

We homogenized the wording to “signal activity” throughout the paper, including the title.

5. No, there is no detectable repeat in the E. coli RPuL2 protein. Instead, we propose in the discussion section (l. 396-403) and in S7 Fig. that basic residues of the RPuL2 proteins interact with surface-exposed acidic residues of the NsrA-receptor protein. 

6. S5 Fig: The figure has now been submitted as a pdf to decrease its size (247Ko). 

7. The abstract was modified as follows to clarify this point “…AOI is initially triggered by so-far unidentified Medicago nodule signals named signal 1 and signal 1’ whose transduction in bacteroids requires the S. meliloti outer-membrane-associated NsrA receptor protein and the cognate inner-membrane-associated adenylate cyclases, CyaK and CyaD1/D2, respectively.”

8. See below

Specific comments:

l. 39: Changed as suggested “Biochemical analyses indicated a peptidic nature for signal 1 and, together with proteomic analyses, a universally conserved Medicago ribosomal protein of the uL2 family was identified as a candidate signal 1.”

l. 42 “signaling activity ex planta” was changed for “signal activity” for clarity.

l. 49 Edited as suggested.

l. 60 We totally agree. We were actually referring to signal identification in symbiotic systems in general, not in the rhizobium-legume symbiosis specifically. To remove this ambiguity, we changed the sentence to “Given the critical importance of these signaling events, the molecular identification of the underlying signals is a major challenge in the field.”

l. 76 corrected

l. 102-103 Changed as follows for clarity “Purified MtRPuL2A (MtrunA17Chr7g0247311) displays a strong signal activity in our ex planta bioassay that…”

l. 111. No other bacteria were examined.

l. 126 Added sentence in Fig 1 legend “ … tested for signal activity in S. meliloti reporter strains GMI12052 (wt, black) and GMI12071 (cyaK,grey).”. 

l. 130 “here” deleted.

l. 136 proteinase K corrected.

l. 184 (legend of Fig. 2). Reporter strains are now specified. 

l. 229 changed as suggested “phylogenetic tree of RPuL2 protein sequences from Medicago truncatula…”

l. 256 changed as suggested “As RPs are very basic proteins…”

l. 269 now explicited “M. sativa bacteria-free NAR (Nodulation in the Absence of Rhizobia)…”

l. 280 . No, MtRPuL2 mainly exists as a ribosome–bound protein in nodules. What our results show is that MtRPuL2 also exists as a free (non-ribosomal) protein in nodules. 

l. 330. The used antibody is now specified in the legend of Fig 7 (l. 342): “The insert shows a Western blot of M. truncatula A17 and Mtnf-ya1 nodules with a human anti-RPL8 (RPuL2) antibody.” The commercial source of the antibody is specified in the Material and Methods section (l. 582).

l. 338 “ a blue coloration” corrected.

l. 341 “Plants were grown under aeroponic conditions.” Corrected

l. 348 Corrected as “Sensing of MtRPuL2A by reporter bacteria has the same genetic requirements as signal 1 sensing…”

l. 350 corrected as suggested “the S. meliloti NsrA receptor protein is involved in recognition of two different signals in symbiosis…”

l. 362: This is a very good point. Yet we think it is unlikely that signaling relates to nodule senescence. Expression of the reporter gene fusion (smc02178-lacZ) in planta takes place in very young (7dpi) and young (14 dpi) mature nodules that show no senescence zone. In mature nodules, expression of the fusion is clearly visible in the infection zone as well as in the early nitrogen-fixing zone (ZIII) of the nodules (see Fig 3 in Tian et al. 2012 PNAS USA 109:6751). This is now stated in the discussion section (l. 364) as follows “Noteworthy, signaling takes place in very young (7dpi) and in the nitrogen-fixing zone (ZIII) of 14dpi nodules (ref Tian et al. 2012), thus making it unlikely that signaling takes place during nodule senescence.”

l. 377 Phosphorylation is one example (ref 42). Now indicated as “Free RPs can be post-translationally modified (eg phosphorylated) or complexed... ”

l. 387 The MDM2 name comes from Mouse double minute 2 homolog. More significantly, it is an ubiquitin ligase. 

l. 410-411. References are now included “ We have detected over the years signal 1 activity in nodules of Medicago plants grown under a variety of conditions (this study, [16,17,46].”

l. 414 Fix- corrected

l. 423 The reference number is now included 

l. 429 Rpf84 corrected. Rpf84 is a RPL22 protein, no homology to RPuL2 proteins. 

l. 469-474 Strain construction used standard published procedures so anyone in the field should be able to follow the construction. “The S. meliloti GMI12072 was constructed by introduction of the plasmid pGD2178 into a nsrA mutant strain (GMI12049,(17) by triparental mating using pRK600 as a helper plasmid. The S. meliloti GMI12071 was constructed by elimination of the gentamycin resistance marker of cyaK in GMI11556 (16) by marker exchange using the sacB selection procedure (56). Next, plasmids pGD2178 and pGMI50333 were introduced by triparental mating using pRK600 as a helper plasmid.” 

We have added the reference to S4 table as requested “Strains genotype is described in S4 Table” (l. 474).

l. 495 spaces included 

l. 586 It is a commercial human antibody. Sigma Prestige Antibodies are tested for no/very low cross reactivity (see HPA050165 notice). There is no described cross reactivity of this antibody with other RPs. 

l. 650 We are still waiting for Genbank accession number. We will communicate it to Plos One as soon as we get it. 

l. -651 Computer programs are indicated “The tree in Fig 3 was computed using https://ngphylogeny.fr (PhyML+SMS) (60), and visualized with Itol (61).”

L 665 The references have been edited to PLOS One style.

Fig 1, 2, 4, 6 Miller units. 

- In most cases (Fig 1BC, 2BCD, 4BC, 6) we used Miller units to compare the activity of the samples to the buffer control. Hence we show that the proteinase K treatment (Fig 1BC), the inactivation of the cyaK or nsrA genes in the reporter strain (Fig 2C, 4B) or the fiberglass treatment (Fig 2D, 4C) all reduced signal activity to buffer (background) levels. Similarly, we showed that the activity of purified ribosomes before RNAse treatment (Fig6) did not exceed background (Buffer) levels. 

- They were two exceptions to this rule. In FIg 1A we compared the signal activity of different biological crude extracts. Then we standardized the activity to the amount (µg) of protein present in the sample. In Fig 5 we tested the activity of purified proteins of very different sizes (ca 1000 to 35 000 Da). Therefore we standardized the activity to the µM concentration of the protein to take into account size differences. 

- The same rationale applies to SI figures. 

Fig. 2 was modified as requested. Fig 2B is a single full purification experiment specifically performed for the publication (ie the three streptag-heparine-gel filtration steps have been performed in a row, no stop, no freezing in between). However, each individual purification step was optimized in preliminary assays. Fig 2C: error bars have been included as requested. Fig 2D is the result of a single experiment (note however that fiber-glass binding was demonstrated for the cognate Medicago protein (Fig 4C).

Fig 3 Panel A the resolution has been improved and the size of the text of the scale bar enhanced, as requested. All public accession numbers are shown. Accession numbers for the Medicago proteins are pending (revised annotation of sequences submitted to GenBank as indicated in the Material and methods section, l.651).

Fig 3B. We do not think it would be wise to include E. coli RPuL2 in this alignment (although it is very easy). As discussed above (reviewer #1), this alignment would reflect the phylogenetic relatedness of the E. coli and Medicago RPuL2 proteins without telling anything about the residues/motif important for signal activity. Instead, we propose in the paper (l. 396-403 and S7Fig) that basic residues of the RPuL2 proteins interact with surface-exposed acidic residues of the NsrA-receptor protein. In S7 Fig (that this reviewer could not review because the file was too large) we show the distribution of basic residues in both E. coli and Medicago RPuL2 proteins.

Fig 4A “kDa” added.

Fig 4BC: Error bars have been added as requested after incorporating a new experiment to the data set. The legend has been modified accordingly. 

Fig5 Error bars have been added as requested. Because the tested peptides/proteins have very different sizes (ca from 1.000 da to 30.000 da), we cannnot express here activity/µg of protein. Activity/µM is better.

Fig 7 “FW” added as suggested in the title of the Y axis.

Fig 7B : the whole western blot picture is now shown. No degradation product was observed. See also the full-length image deposited at Plos One site. 

End.

---

## [Decision Letter · Decision Letter 1]

21 May 2020

PONE-D-20-05716R1

The ex planta signal activity of a Medicago ribosomal uL2 protein suggests a moonlighting role in controlling secondary rhizobial infection

PLOS ONE

Dear Dr. %Batut%,

Thank you for submitting your new revision of your manuscript to PLOS ONE. One of the reviewers still have some minor points to clarify in this new version. Therefore, we invite you to submit a revised version of the manuscript that addresses these particular points.

We look forward to receiving your revised manuscript.

Kind regards,

Francisco Martinez-Abarca, Ph.D.

Academic Editor

PLOS ONE

Reviewers' comments:

Reviewer's Responses to Questions

**Comments to the Author**

1. If the authors have adequately addressed your comments raised in a previous round of review and you feel that this manuscript is now acceptable for publication, you may indicate that here to bypass the “Comments to the Author” section, enter your conflict of interest statement in the “Confidential to Editor” section, and submit your "Accept" recommendation.

Reviewer #1: All comments have been addressed

Reviewer #2: All comments have been addressed

2. Is the manuscript technically sound, and do the data support the conclusions?

Reviewer #1: Yes

Reviewer #2: Yes

3. Has the statistical analysis been performed appropriately and rigorously? 

Reviewer #1: Yes

Reviewer #2: Yes

4. Have the authors made all data underlying the findings in their manuscript fully available?

Reviewer #1: Yes

Reviewer #2: (No Response)

5. Is the manuscript presented in an intelligible fashion and written in standard English?

Reviewer #1: Yes

Reviewer #2: Yes

6. Review Comments to the Author

Reviewer #1: The authors have addressed satisfactorily most of my minor comments. However, in some of the Figures, I have still detected some small mistakes that should be corrected to reach the quality required for a publication in PLoS One.

-Fig. 1: Some inconsistency is detected in this figure, to which I tried to draw attention with one of my questions. Thus, to the question that I raised “Why was not the cyaK mutant included in the assays shown in Fig. 1B?”, Sorroche et al answered that cyaK-dependency of the Medicago nodule signal was established in Fig 1A. But then, why was the cyaK mutant included in the assays shown in Fig. 1C? cyaK-dependency of the signal present in E. coli extracts was also established in Fig. 1A. Therefore, the authors should be consistent by showing or not cyaK-dependency in both figures Fig.1B and 1C. Likewise, why was not the buffer control included in Fig. 1B like the authors did in Fig. 1C? Furthermore, in the figure legend the amount of E. coli crude extract used to obtain the data shown in Fig. 1C is mentioned, but not that of the nodule extract used for the data shown in Fig. 1B.

-Fig. 2: The authors have forgotten to include “IV” in Fig. 2A and 2B. For Fig. 2B, I think that it would have been more correct to express the data as Units per microgram of protein since the amount of protein after the dilutions of the different fractions might still be different. No error bars have been included in Fig. 2B and 2D, probably because the authors have only one data. To me, this is OK but it should be mentioned (for example in the figure legend). The legend of this figure requires additional minor corrections: Line 178, after “overexpressing strain” include a colon and II like this: “overexpressing strain, II: Pool of…” Moreover, include the amount of protein used in the Fiberglass assay, as you did in the legend of Fig. 4C.

-Fig. 4: Lettering of the X axis in Fig. 4C has disappeared in the new version of this Figure.

-Lines 240-243: The authors compare the specific activity of purified MtRPuL2A and that of E. coli RPuL2 and they conclude that in the former it is ca. 3-fold lower. Was this conclusion obtained by comparing data shown in Fig. 2C and Fig. 4B, in which Miller Units and not Miller Units/ microgr of protein are indicated? Did they use the same amount of protein for the two assays? Please, revise and clarify.

Reviewer #2: The manuscript has been improved. I don't have any further comments. I therefore recommend acceptance of this manuscript.

7. PLOS authors have the option to publish the peer review history of their article (what does this mean?). If published, this will include your full peer review and any attached files.

Reviewer #1: No

Reviewer #2: No

---

## [Author Response · Author response to Decision Letter 1]

12 Jun 2020

Reviewer#2

Fig 1. To improve consistency, we have removed the cyaK and Buffer controls from panel C. 

The amount of nodule extract (50 mg) is now indicated in the legend of Fig. 1B.

Fig. 2. 

- “IV “was added back to panels A and B.

- Miller units is correct here. As shown in panel A, fractions IV 7 to 10 from the size exclusion column contain different amounts of RPul2 protein (which is the major protein in the sample at this stage of purification). We observed that the signal activity (Miller units, panel B) varies with the amount of RPuL2 protein present in the fractions (panel A). 

- The fact that Fig 2B and 2D feature a single assay is now acknowledged at the end of the figure legend (l. 186): “ Panels B and D feature the results of a single typical experiment.”

- Fig 2D: The amount of purified RPuL2 protein used in this assay (1µg) is now indicated in the legend of fig. 2D (l. 185).

Fig 4C. Sorry, the figure was oversized. This has been corrected. 

l.240-243. The specific activity of the E. coli RPuL2 protein was assessed independently in separate experiments, including fig 2C . This is now clarified l. 241 “ The specific activity of the purified MtRPuL2A-strep protein was ca. 3-fold lower than that of E. coli RPuL2 (ca 105 Miller units/µM) assessed in independent purification assays,…”

End.

---

## [Editor Report · Decision Letter 2]

16 Jun 2020

The ex planta signal activity of a Medicago ribosomal uL2 protein suggests a moonlighting role in controlling secondary rhizobial infection

PONE-D-20-05716R2

Dear Dr. Batut,

We’re pleased to inform you that your manuscript has been judged scientifically suitable for publication and will be formally accepted for publication once it meets all outstanding technical requirements.

Kind regards,

Francisco Martinez-Abarca, Ph.D.

Academic Editor

PLOS ONE
---

## [Editor Report · Acceptance letter]

14 Sep 2020

PONE-D-20-05716R2 

The ex planta signal activity of a Medicago ribosomal uL2 protein suggests a moonlighting role in controlling secondary rhizobial infection 

Dear Dr. Batut:

I'm pleased to inform you that your manuscript has been deemed suitable for publication in PLOS ONE. Congratulations! Your manuscript is now with our production department. 

Kind regards, 

on behalf of

Dr. Francisco Martinez-Abarca 

Academic Editor

PLOS ONE